# The Role of ApoE Serum Levels and *ApoE* Gene Polymorphisms in Patients with Laryngeal Squamous Cell Carcinoma

**DOI:** 10.3390/biom12081013

**Published:** 2022-07-22

**Authors:** Rasa Liutkeviciene, Justina Auzelyte, Vykintas Liutkevicius, Alvita Vilkeviciute, Greta Gedvilaite, Paulius Vaiciulis, Virgilijus Uloza

**Affiliations:** 1Neuroscience Institute, Lithuanian University of Health Sciences, LT-44307 Kaunas, Lithuania; rasa.liutkeviciene@lsmuni.lt (R.L.); alvita.vilkeviciute@lsmu.lt (A.V.); greta.gedvilaite@lsmu.lt (G.G.); 2Medical Academy, Lithuanian University of Health Sciences, LT-44307 Kaunas, Lithuania; justina.auzelyte@stud.lsmu.lt; 3Department of Otorhinolaryngology, Lithuanian University of Health Sciences, LT-44307 Kaunas, Lithuania; vykintas.liutkevicius@lsmuni.lt (V.L.); virgilijus.ulozas@lsmuni.lt (V.U.)

**Keywords:** ApoE, laryngeal squamous cell carcinoma, rs7412 and rs429358, survival rate

## Abstract

Recent studies have revealed that the inflammatory *ApoE* effect may play a significant role in various cancer development. However, this effect has still not been analyzed in patients with laryngeal squamous cell carcinoma (LSCC). In the present study, we evaluated two single nucleotide polymorphisms (SNPs) of *ApoE* (rs7412 and rs429358) and determined their associations with LSCC development and the LSCC patients’ five-year survival rate. Additionally, we analyzed serum *ApoE* levels using an enzyme-linked immunosorbent assay. A total of 602 subjects (291 histologically verified LSCC patients and 311 healthy controls) were involved in this study. The genotyping was carried out using the real-time PCR. We revealed that *ApoE* ε3/ε3 was associated with a 1.7-fold higher probability of developing LSCC (*p* = 0.001), with 1.7-fold increased odds of developing LSCC without metastasis to the lymph nodes (*p* = 0.002) and with a 2.0-fold increased odds of developing well-differentiated LSCC (*p* = 0.008), as well as 1.6-fold increased odds of developing poorly differentiated LSCC development (*p* = 0.012). The *ApoE* ε2/ε4 and ε3/ε4 genotypes were associated with a 2.9-fold and 1.5-fold decrease in the likelihood of developing LSCC (*p* = 0.042; *p* = 0.037, respectively). *ApoE* ε3/ε4 was found associated with a 2.4-fold decreased likelihood of developing well-differentiated LSCC (*p* = 0.013). Conclusion: *ApoE* ε2/ε4 and ε3/ε4 were found to play a protective role in LSCC development, while *ApoE* ε3/ε3 may have a risk position in LSCC development.

## 1. Introduction

Laryngeal squamous cell carcinoma (LSCC) is one of the most common tumors of the head and neck region malignancies [1]. LSCC is more often diagnosed in men than women, mainly between the ages of 50 and 54 (in Europe) and between the ages of 75 and 79 (in Asia) [2,3,4,5]. The etiology of LSCC is a combination of different factors such as usage of tobacco products, alcohol consumption, human papillomavirus infection, long-term exposure to various environmental chemical compounds, previous radiation, and genetic factors [6]. It is also important to note that LSCC causes a variety of symptoms, which may manifest differently depending on the laryngeal site affected by the tumor, such as voice changes or difficulty swallowing, chronic cough, dyspnea, and fatigue [7,8]. According to the Global Health Data Exchange, 1.66 per 100,000 inhabitants died from the LSCC in 2020 worldwide. Additionally, the mortality rate increases 5-fold if the patient is male and older (>65 years) [9]. In the Lithuanian population, LSCC caused over 120 deaths in 2020 [5]. Despite the improving diagnostic techniques of LSCC, the overall survival rate (OS) remains low, especially in older and comorbid patients [10,11]. These data encourage to conduct of further research on potential blood- or tissue-based biomarkers for early LSCC diagnosis. Intensive and long-lasting smoking and alcohol consumption as a potentialyzer for cancerogenic smoking are the potential risk factors for the development of LSCC and are also influenced by some genetic factors [12,13,14]. The genetic factor that could potentially be important in the LSCC development is the *ApoE* gene. The *ApoE* gene is located on chromosome 19 (19q13.32) and encodes ApoE, a plasma protein that serves as a ligand for low-density lipoprotein receptors [15]. This protein is secreted by many different tissue types, such as hepatocytes (up to 75%), and also by adipose tissue, kidneys, adrenal glands, and the brain [16]. Moreover, *ApoE*, as a compound of protein and lipid, carries out many functions and is particularly involved in lipid metabolism and homeostasis. It also impacts tumorigenesis, including proliferation, angiogenesis, and metastasis [17,18]. In humans, there are three isoforms of the ApoE protein: ApoE ε2 (Cys112, Cys158), ε3 (Cys112, Arg158), and ε4 (Arg112, Arg158) [19] and encoded by *ApoE*’s ε2/ε3/ε4 allelic variants, which are determined by two single nucleotide polymorphisms (SNPs, rs429358 and rs7412): the rs7412 (T) allele indicates the presence of ε2 allele (Cys112, Cys158), whereas the C allele together with the common rs429358 (T) allele defines the ε3 allele (Cys112, Arg158). If the rs429358 (C) allele is found together with the rs7412 (C) allele, the combination is known as an APOE-ε4 allele (Arg112, Arg158) [20].

Although the mechanisms of action are still unclear, the *ApoE* gene is involved in dementia, Alzheimer’s disease (AD), Parkinson’s disease, and cardiovascular disease (ischemic stroke) development [21].

*ApoE* isoforms, especially *ApoE* ε4, are involved in inflammatory processes in microglia and astrocytes. Being aware that inflammatory processes are also involved in carcinogenesis [22], it can be hypothesized that *ApoE* could potentially influence cancer development. Other factors that can influence inflammation are estrogen receptors (ER), which are also registered to be associated with *ApoE* and tumorigenesis [23,24,25]. It is also important to note that ER are modulators of the LSCC tumorigenesis and may be used as potential biomarkers in the diagnostics of this tumor [26]. Since macrophages produce apolipoproteins, including *ApoE*, it can be argued that macrophages secrete *ApoE* in cancerous tissues. *ApoE* expression in cancerous tissues has a bigger potential for tissue invasion and metastasis, which can be associated with poor OS [17,18,27].

Since *ApoE* is involved in the development of many different diseases, the matter of question is how *ApoE* can be associated with cancerous processes. It has been previously reported that patients with the *ApoE* ε4 allele showed a lower chance of melanoma progression and metastasis compared to patients with the *ApoE* ε2 allele [28]. Moreover, a recent study has shown that activation of the axial mechanism of liver-X nuclear receptor (LXR) and *ApoE* by pharmacological agents can elicit an immune response against a variety of cancers, such as glioblastoma, ovarian, renal, colon cancers, and lung cancer [29]. Another study has found that IL3 (one of the cytokines), acting in combination with *ApoE*, promotes the migration and angiogenesis of non-small-cell lung cancer (NSCLC) cells, thus causing metastasis [30]. It has also been identified that NSCLC patients have elevated levels of ApoE in the bronchial walls [31]. All of these data suggest that high levels of *ApoE* are associated with lung cancer. In contrast, some studies have shown that NSCLC is more associated with a risk of metastasis in patients without the *ApoE* gene [32]. Not only is *ApoE* a potential biomarker for lung cancer, but it has also been reported to be one of the biomarkers to help diagnose breast cancer in men [33]. *ApoE* is associated with an increased risk of developing various cancers, including the nasopharynx [18,32,33,34]. However, the interactions between *ApoE* and LSCC have not been adequately explored.

The present study aimed to determine the associations of single nucleotide polymorphisms (SNPs) of the *ApoE* gene and *ApoE* serum levels with clinical and morphologic manifestations of LSCC and the 5-year survival of the patients.

## 2. Materials and Methods

This case-control study was conducted at the Department of Otorhinolaryngology, Lithuanian University of Health Sciences (LUHS), Kaunas, Lithuania, 2009–2020. The research protocol was approved by the Kaunas Regional Biomedical Research Ethics Committee (BE-2-37). All procedures performed in the study conformed to the institution’s ethical standards, the Declaration of Helsinki and its subsequent amendments, or similar ethical standards. Informed consent was obtained from all subjects participating in the study.

### 2.1. Study Protocol/Design

**Study Population**. The total study group consisted of 602 subjects: 291 patients with LSCC and 311 healthy subjects as a control group. The LSCC patient group included 282 (96.9%) men and 9 (3.1%) women with a mean age of 64 (interquartile range (IQR) 9) years. The control group included 301 (96.8%) men and 10 (3.2%) women with a mean age of 66 (IQR 9) years. The LSCC patients and healthy control groups were adjusted for sex and age.

**LSCC group.** The LSCC group consisted of glottis region LSCC patients. All LSCC patients, before the treatment, underwent a detailed otolaryngologic examination, including flexible endoscopy and/or video laryngostroboscopy, in the Department of Otorhinolaryngology, LUHS. Peripheral venous blood samples were obtained before induction of general anesthesia and direct microlaryngoscopy with biopsy. The diagnosis of LSCC was confirmed histologically in the Department of Pathology, LUHS. The final diagnosis of LSCC was based on the clinical data and the results of histological examination, as well as the laryngeal and neck data on computed tomography or magnetic resonance imaging. The staging of LSCC was performed according to the description of the American Joint Committee on Cancer [35].

**Healthy controls**. Patients who were consulted in the Department of Otorhinolaryngology, LUHS, and scheduled for surgical treatment (tympanoplasty, ossiculoplasty, tympanostomy, nasal bone reposition septoplasty, rhinoseptoplasty, uvulopalatopharyngoplasty, or radiofrequency thermoablation of the soft palate) were enrolled into the present study. Peripheral venous blood samples from these patients were obtained through the same catheter used for induction of general anesthesia. Additionally, patients without previously diagnosed oncologic disease who came to the primary care physician’s office for a general health examination and had a peripheral blood sample collection were also included in this study as healthy controls. All patients diagnosed with any other type and location of cancer, acute or chronic infectious disease, subjects taking psychomotor suppressants and antiepileptic drugs, and subjects younger than 18 years of age were excluded from this study.

**Survival Rate**. Data on the mortality rate of the LSCC group, including survival after diagnosis of LSCC and cause of death, were obtained from the Lithuanian State Registry of Deaths and Their Causes.

### 2.2. DNA Extraction, ApoE Genotyping, and Enzyme-Linked Immunosorbent Assay

Peripheral venous blood samples from LSCC patients and controls were collected in ethylenediaminetetraacetic acid (EDTA)-containing vacutainer tubes and stored immediately at −20 °C until further analysis. Deoxyribonucleic acid (DNA) was extracted from peripheral blood leukocytes using a reagent kit (NucleoSpin Blood L Kit; Macherey & Nagel, Düren, Germany). Before a real-time polymerase chain reaction (RT-PCR), the concentration of DNA was measured using a spectrophotometer (Agilent Technologies, Penang, Malaysia). Two SNPs of the *ApoE* gene (rs429358 and rs7412) were identified using the commercially available genotyping kits C_084793_20 and C_904973_10. (Applied Biosystems, Foster City, CA, USA). DNA samples were set out in a 96-well plate layout before genotyping. A total of 95 wells on the plate were filled with individual DNA samples. One empty well (no template control, NTC) was used to reveal potential contaminations. We used the Applied Biosystems 7900HT Real-Time Polymerase Chain Reaction System to detect SNPs. The cycling program began with heating at 95 °C for 10 min, followed by 40 cycles of heating at 95 °C for 15 s and 60 °C for 1 min. Allele discrimination was performed using Applied Biosystems software (Applied Biosystems, Carlsbad, CA, USA, version 2.3). Serum APOE levels were determined in 18 control subjects and 20 LSCC patients. Serum APOE levels in LSCC patients were determined using a commercially available enzyme-linked immunosorbent assay (ELISA) kit for human APOE (APOE (AD2) ELISA Kit, Abcam, Cambridge, UK). According to the manufacturer’s instructions, optical density was measured immediately at a wavelength of 450 nm using a microplate reader (Multiskan FC microplate photometer, Thermo Scientific, Waltham, MA, USA). The APOE level was calculated using the standard curve; the sensitivity range of the standard curve: 0.63–40 ng/mL, sensitivity 132 ng/mL.

### 2.3. Statistical Analysis

Statistical analysis was performed using SPSS/W 27.0 software (Statistical Package for the Social Sciences for Windows, Inc., Chicago, IL, USA). Data were expressed as absolute numbers with percentages. Genotype frequencies were expressed as percentages. Hardy–Weinberg analysis compared the observed and expected frequencies of *ApoE* genotypes with the χ^2^ test in all groups. The distribution of *ApoE* genotypes in LSCC and control groups was compared using the χ^2^-test. The non-parametric Mann–Whitney U test was used to compare age between two groups. Binary logistic regression analysis was performed to estimate the impact of genotypes on LSCC development. Odds ratios and 95% confidence intervals are presented. The selection of the best genetic model was based on the Akaike Information Criterion (AIC); therefore, the best genetic models were those with the lowest AIC values. Survival curves were estimated by the Kaplan–Meier method, and median survival time was reported with a 95% confidence interval (95% CI). The log-rank test was used to determine if there was a difference in survival curves between different groups of patients. Differences were considered statistically significant when *p* < 0.05

## 3. Results

The demographic characteristics of the study groups are shown in Table 1. The distributions of *ApoE* (rs7412 and rs429358) gene polymorphisms in the control group were in accordance with Hardy—Weinberg equilibrium (HWE) (*p* > 0.001) (data not shown).

We analyzed the frequency distributions of the *ApoE* genotypes and alleles in LSCC and control groups. Statistical analysis revealed significantly lower frequency of *ApoE* ε2/ε4 and ε3/ε4 genotypes and ε4 allele in the group of LSCC patients compared to the control group (1.7% vs. 4.8% *p* = 0.034, 18.9% vs. 26.0% *p* = 0.001, 13.9% vs. 20.6% *p* = 0.002, respectively). Moreover, the frequency of *ApoE* ε3/ε3 genotype and ε3 allele was significantly higher in the LSCC patient group compared to the control group (61.2% vs. 48.2% *p* = 0.001, 77.5% vs. 68.8% *p* < 0.001, respectively) (Table 2).

Binary logistic regression analysis was performed to evaluate the impact of *ApoE* on the development of LSCC. We found that *ApoE* ε2/ε4 and ε3/ε4 genotypes were associated with 2.9-fold and 1.5-fold decreased odds of LSCC development (OR = 0.345; CI: 0.124–0.962; *p* = 0.042; OR = 0.662; CI: 0.449–0.975; *p* = 0.037, respectively), while *ApoE* ε3/ε3 genotype was associated with a 1.7-fold increased odds of LSCC development (OR = 1.691; CI: 1.223–2.338; *p* = 0.001). The results are shown in Table 3.

The patient group was divided into subgroups according to the stage of the disease: early stage (I–II stage) and advanced stage (III–IV stage). Statistical analysis revealed statistically significant differences between *ApoE* ε3/ε3 genotype ε3 and ε4 allele in both the group of patients with LSCC in early and advanced stages compared with the control group. *ApoE* ε3/ε3 was more frequent in the LSCC patients in the early stage than in the control group (60.1% vs. 48.2% *p* = 0.013) and in the advanced stage (62.6% vs. 48.2% *p* = 0.007, respectively). The ε3 allele was more frequent in the group of LSCC patients in the early and advanced stages than in the control group (75.9% vs. 68.8% *p* = 0.021 and 77.1% vs. 68.8% *p* = 0.019, respectively). In contrast, the ε4 allele was less frequent in the early and advanced stage LSCC groups than in the control group (15.2% vs. 20.6% *p* = 0.041, 13.9% vs. 20.6% *p* = 0.029, respectively) (Table 4).

Results of binary logistic regression showed that the *ApoE* ε3/ε3 genotype was associated with 1.6-fold increased odds of early LSCC development (OR = 1.618 95% CI = 1.106–2.367; *p* = 0.013). It was also associated with 1.8-fold increased odds of advanced stage LSCC development (OR = 1.797 95% CI = 1.172–2.755; *p* = 0.007) (Table 5).

The group of LSCC patients was divided into four subgroups according to the tumor size: T1, T2, T3, and T4. *ApoE* genotypes and allele frequencies were analyzed between the subgroups of the patients and the control groups. It was found that the *ApoE* ε3/ε3 genotype was more frequent in the T1 subgroup than in the control group (61.7% vs. 48.2% *p* = 0.016), while the *ApoE* ε3/ε4 genotype and the ε4 allele were less frequent in the T1 subgroup than in the control group (14.0% vs. 26.0%, *p* = 0.011; 14.0% vs. 20.6%, *p* = 0.034, respectively). We also found that the ε3 allele was more common in the T3 subgroup than in the control group (77.9% vs. 68.8% *p* = 0.045). Analysis of the T4 subgroup compared to the control group revealed that the *ApoE* ε3/ε3 genotype and the ε3 allele were more frequent in the T4 subgroup than in the control group (63.5% vs. 48.2%, *p* = 0.027; 79.4% vs. 68.8%, *p* = 0.018, respectively), while the ε4 allele was less frequent in the T4 group than in the control group (11.9% vs. 20.6%, *p* = 0.024) (Table 6).

Binary logistic regression was performed to evaluate the influence of *ApoE* on the development of LSCC with different tumor sizes. The analysis revealed that *ApoE* ε3/ε3 genotype was associated with 1.7-fold increased odds of developing LSCC with T1 tumor size (OR = 1.728 95% CI = 1.103–2.706; *p* = 0.017), while ε3/ε4 was associated with 2.2-fold decreased odds of LSCC with T1 tumor size development (OR = 0.463 95% CI = 0.254–0.845; *p* = 0.012). *ApoE* ε3/ε3 genotype was also associated with 1.9-fold increased odds of developing LSCC T4 tumor (OR = 1.867 95% CI = 1.067–3.267; *p* = 0.029) (Table 7).

We analyzed the distributions of *ApoE* genotypes and allele frequencies between the control group and LSCC patients with metastases (N1–2) and without (N0) metastases to the neck lymph nodes. The frequency analysis of *ApoE* genotypes and alleles revealed that *ApoE* ε3/ε3 genotype and ε3 allele were more frequent in the LSCC (N0) group than in the control group (61.2% vs. 48.2%, *p* = 0.002; 77.2% vs. 68.8%, *p* = 0.002, respectively). At the same time, the ε4 allele was less frequent in LSCC (N0) subgroup than in the control group (14.9% vs. 20.6%, *p* = 0.014). The *ApoE* ε2/ε2 genotype was more common in the LSCC (N1–2) subgroup than in the control group (4.3% vs. 0.6%, *p* = 0.026). In contrast, the ε4 allele was less common in the LSCC (N1–2) subgroup than in the control group (9.8% vs. 20.6%, *p* = 0.014) (Table 8).

Binary logistic regression revealed that *ApoE* ε3/ε3 genotype was associated with 1.7-fold increased odds of LSCC (N0) (OR = 1.695 95% CI = 1.206–2.381; *p* = 0.002) (Table 9).

The LSCC patient group was divided into two subgroups: well-differentiated (G1) and poorly differentiated (G2-G3) LSCC. We analyzed the distributions of *ApoE* genotypes and allele frequencies in the control group and patients with LSCC subgroups according to the tumor differentiation grades. The analysis showed that *ApoE* ε3/ε3 genotype and ε3 allele were more frequent in the subgroup of well-differentiated LSCC than in the control group (64.7% vs. 48.2% *p* = 0.007; 78.8% vs. 68.8% *p* = 0.011, respectively), while *ApoE* ε3/ε4 genotype and ε4 allele were less frequent (12.9% vs. 26.0% *p* = 0.011, 11.2% vs. 20.6% *p* = 0.005, respectively). *ApoE* ε2/ε4 genotype and ε4 allele were less frequent in the poorly differentiated LSCC subgroup than in the control group (1.5% vs. 4.8% *p* = 0.042, 15.4% vs. 20.6% *p* = 0.035, respectively), while *ApoE* ε3/ε3 genotype and ε3 allele were more frequent (59.5% vs. 48.2% *p* = 0.012, 76.6% vs. 68.8% *p* = 0.007, respectively) (Table 10).

Binary logistic regression revealed that *ApoE* ε3/ε3 and ε3/ε4 have an opposite effect on the development of LSCC in terms of the degree of tumor differentiation. *ApoE* ε3/ε3 is likely associated with 2.0-fold increased odds of well-differentiated LSCC development. While *ApoE* ε3/ε4 is likely associated with 2.4-fold decreased odds of well-differentiated LSCC development (OR = 1.968 95% CI = 1.197–3.236; *p* = 0.008, OR = 0.422 95% CI = 0.213–0.835 *p* = 0.013, respectively) (Table 11). Additionally, *ApoE* ε3/ε3 is likely associated with 1.6-fold increased odds of poorly differentiated LSCC development (OR = 1.578 95% CI = 1.104–2.254; *p* = 0.012) (Table 11).

Twenty random serum samples from the LSCC group were selected for the APOE concentration measurement. The Control group consisted of 18 subjects considering the age and gender distributions based on the LSCC group. The assay was performed in duplicates for all study samples.

The statistical analysis did not show statistically significant differences between the groups (1.8 ng/mL ± 0.347 vs. 2.01 ng/mL ± 0.440; *p* = 0.113) (Figure 1).

We also compared the genotypic distribution of ApoE serum levels between the control and LSCC groups. No statistically significant differences were found (Table 12).

The specific disease 5-year survival rate of LSCC patients was 67.4%. When we analyzed the 5-year survival rate of LSCC patients and the distribution of *ApoE* genotypes (ε1ε3, ε1ε4, ε2ε2, ε2ε3, ε2ε4, ε3ε3, ε3ε4, and ε4ε4), we found no statistically significant effect of these *ApoE* genotypes on the 5-year survival rate of the patients (Figure 2 and Figure 3).

## 4. Discussion

The relation between the *ApoE* gene polymorphisms and LSCC has not been sufficiently covered in the literature [36]. In the present study, we investigated the potential involvement of *ApoE* gene polymorphisms in LSCC tumorigenesis and their potential to serve as diagnostic and prognostic molecular markers. This study is the first to investigate *ApoE* SNPs and serum ApoE levels in a pure and homogeneous cohort of patients with LSCC. To the best of our knowledge, no studies analyzing the impact of ApoE serum levels and *ApoE* gene polymorphisms in patients with LSCC have been carried out. Previous studies on the morphogenesis of cancer have drawn attention to the role of ApoEs, especially in the development of head and neck cancer and other types of cancers. In our study, we proved that *ApoE* ε2/ε4 and ε3/ε4 were found to play a significant role in LSCC development, while *ApoE* ε3/ε3 may have a protection position in LSCC development. No association was found between ApoE serum levels and LSCC development.

*ApoE* consists of 299 amino acids with numerous amphipathic *α*-helices. *ApoE* has three main alleles: *ApoE*-ε2 (cys112, cys158), *ApoE*-ε3 (cys112, arg158), and *ApoE*-ε4 (arg112, arg158). These allelic forms differ from each other by two amino acids at positions 112 and 158; these differences alter *ApoE* structure and function [37]. *ApoE* has many functions besides its well-known role in lipid metabolism, which is potentially involved in cancer risk. Indeed, Trompet et al. addressed the relationship between cholesterol and cancer by studying both the effect of plasma cholesterol levels and *ApoE* isoforms on overall cancer risk during a 3-year follow-up study in a large elderly cohort. Results showed an inverse relationship between cancer incidence or mortality and cholesterol levels, whereas no effect was shown for the *ApoE* alleles [38]. *ApoE* has also been shown to be involved in tissue repair, inflammatory and immune response, cell growth, and angiogenesis [39] and also shows antioxidant properties [40]. Moreover, *ApoE* is upregulated and exerts biological functions in various cancers [41]. Activation of ApoE restricted the innate immune system’s suppression of cancer cell proliferation, thus promoting tumor growth and metastasis in many types of cancers [29]. The functions of ApoE in DNA synthesis, cell proliferation, and angiogenesis have been identified, such that disruption of these functions potentially leads to tumorigenesis and progression [27]. However, the exact mechanism underlying the effect of ApoE on tumorigenesis is still unknown. ApoE might strongly inhibit the proliferation of various cell types and influence angiogenesis, tumor cell growth, and metastasis by modulating the production of cytokines, growth factors, and other molecules [29].

The effect of *ApoE* genotypes has been investigated concerning breast, colorectal, biliary tract, prostate cancer, and hematologic malignancies with conflicting results [42,43,44,45]. ApoE is significantly highly expressed in lung cancer, and its overexpression promotes cancer proliferation and migration and aggressiveness of lung cancer [46]. ApoE knockout inhibits tumor growth and metastasis by increasing REEM-1-mediated infiltration of natural killer cells in lung cancer [46]. Sakashita et al. investigated the relationship between ApoE and gastric cancer by immunohistochemistry and RT-PCR, and ApoE overexpression in gastric cancer tissues exhibited stronger malignant invasiveness compared to cancer tissues with low ApoE expression [47]. ApoE was upregulated in gastric cancer, and such patients had shorter survival times. There was a strong link between ApoE levels and the risk of muscular invasion, making it a promising marker for predicting the invasions of gastric tumors [47,48], and overexpression of ApoE significantly promoted the abilities of invasion and lymph node metastasis of gastric cancer cells [47]. ApoE was overexpressed in various ovarian cell lines and tissues, and it was essential for the growth and survival of ovarian cancer cells [49]. The level of ApoE in the serum of patients with ovarian cancer was dramatically increased over healthy individuals, and as a marker, it could enhance the specificity and sensitivity of ovarian cancer diagnosis [50]. ApoE was highly expressed in the PC-3 human prostate cancer cell line, and its expression was directly correlated with the Gleason score of prostate cancer tissues, hormone independence, and local and distant metastasis [51]. Additionally, the higher ApoE levels correlated with lymph node and distant metastasis, TNM stages, and poor prognosis in NSCLC patients [52]. Interestingly, higher ApoE protein levels appear to have a suppressive effect on melanoma invasion and metastasis [53].

There is some scientific evidence that *ApoE* affects tumor development, including head and neck tumors [36,54]. Lin et al.’s study found that *ApoE* and regulation of apoptosis by parathyroid hormone-related proteins are significantly associated with ferroptosis and immune cells in papillary thyroid carcinoma [54]. Other scientists’ study group provided novel evidence of a possible protective effect of the ε2 allele against head and neck cancer, probably due to its increased antioxidant properties [36]. Xiao et al. found that the *ApoE* rs429358-TC genotype showed an elevated risk of developing thyroid cancer (*p* = 0.002), whereas *ApoE* rs7412-CT/TT was a protective genotype against the risk of this disease (*p* = 0.0003) [55]. Xue et al. investigated the links between *ApoE* and nasopharyngeal carcinoma. According to that study, patients with nasopharyngeal carcinoma had a statistically significant increase in serum *ApoE* levels along with statistically significantly (*p* < 0.05) more common metastases to the lymph nodes and invasion of surrounding tissues [34]. These findings are in concordance with Jaykar et al.’s statement that reducing the amount of *ApoE* reduces the invasion of tumor cells into healthy tissues by minimizing matrix degradation and the number of invadopodia [56].

Based on the results from the present study, carriers of the ε2/ε4 and ε3/ε4 *ApoE* genotypes had a statistically significant decreased probability of LSCC development. Furthermore, patients with these genotypes had a 2.9- and 1.5-fold decreased risk of developing LSCC, respectively. Conversely, the *ApoE* ε3/ε3 genotype was associated with a 1.7-fold increased risk of developing LSCC. However, in the present study, we did not observe significant differences in serum ApoE levels between the LSCC and control groups. No correlation was identified between ApoE protein levels and the 5-year survival rate of patients with LSCC. No effect of ApoE SNPs on the 5-year survival rate of these patients was revealed.

The strength of the present study was the involvement of a large study population (total, 602 subjects, including 291 LSCC patients and 311 controls), adjustment of patient and control groups for age and sex, and enrolment of pure LSCC patients’ cohort with tumor localization in glottis region. These features of the study ensured a comprehensive analysis of the associations between the selected *ApoE* gene SNPs, *ApoE* serum levels and clinical, morphologic manifestations, and 5-year survival of the patients of this specific tumor in one anatomical region of the head and neck (i.e., LSCC). Most of the studies presented in the literature provide results of genetic analysis unified under the umbrella of the head and neck squamous cell carcinoma (HNSCC) term, which covers malignant tumors of different localizations (oral, pharyngeal, nasopharyngeal, hypopharyngeal, laryngeal regions, etc.), diverse etiology, biological and clinical behavior [57]. Therefore, the pooling of different cancer types into one cohort may mask possible significant associations of selected biomarkers with individual cancer types. The LSCC is manifesting as a less aggressive tumor, assuming a rather low metastatic rate and local spreading compared to other HNSCCs [58]. Therefore, the results of the present study showing the decreased odds of LSCC development in *ApoE* ε2/ε4 and ε3/ε4 genotypes carriers as well as increased odds in *ApoE* ε3/ε3 genotype carriers, point out the importance of these SNPs and haplotypes in the LSCC pathology.

Several limitations of the present study must be considered. The sample size for analysis of ApoE serum concentrations levels was rather limited and too small to reach the desired power setting. Therefore, the results obtained in the present study regarding the associations between the ApoE serum concentration levels and LSCC should be considered as the tendency. Further investigation with a large enough sample size is advocated to confirm the possible role of serum concentration levels of ApoE in LSCC development. Furthermore, the analysis of important etiological factors in LSCC development, i.e., smoking and alcohol consumption, was not carried out in the present study. However, this is foreseen as a targeted task in future investigations. In the present study, all genetic investigations were performed at the baseline before the treatment, i.e., immediately after diagnosing LSCC. Analysis of the potential effect of different LSCC therapy regimens will be foreseen for further future studies.

## 5. Conclusions

*ApoE* ε2/ε4 and ε3/ε4 were found to play a protective role in LSCC development, while ApoE ε3/ε3 may have a risk position in LSCC development.

## Figures and Tables

**Figure 1 biomolecules-12-01013-f001:**
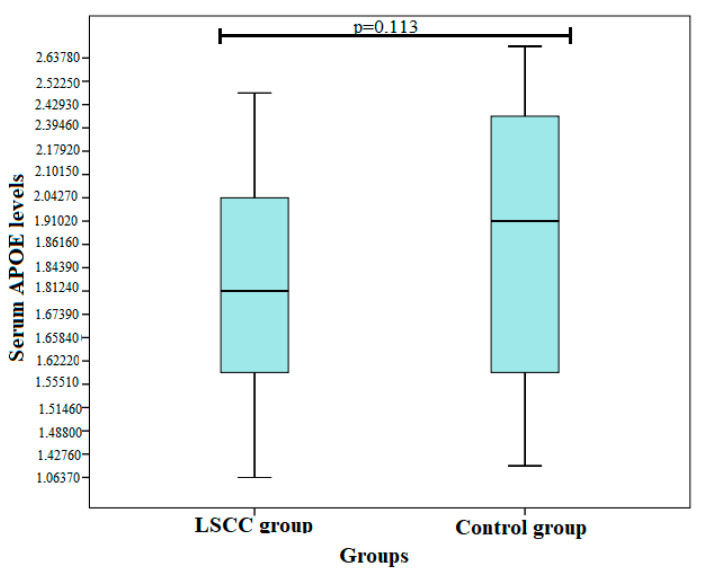
Serum ApoE levels in LSCC and control groups.

**Figure 2 biomolecules-12-01013-f002:**
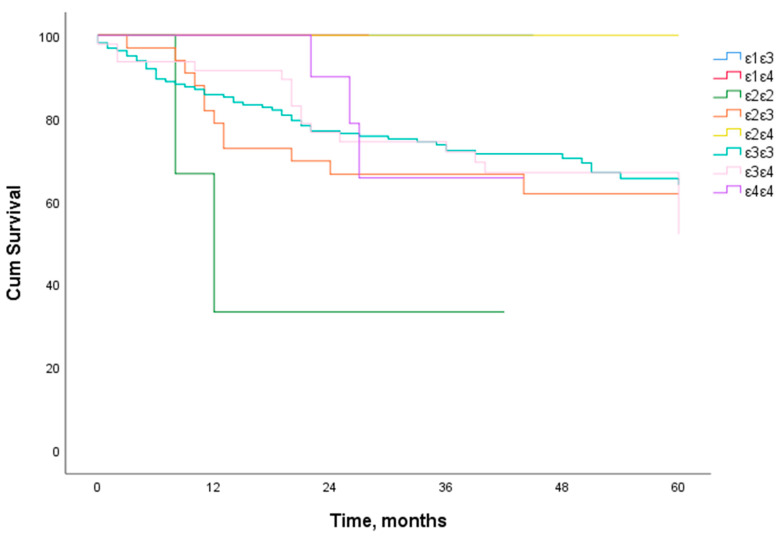
Five-year specific disease survival analysis of LSCC patients according to *ApoE* genotypes (ε1ε3, ε1ε4, ε2ε2, ε2ε3, ε2ε4, ε3ε3, ε3ε4, and ε4ε4).

**Figure 3 biomolecules-12-01013-f003:**
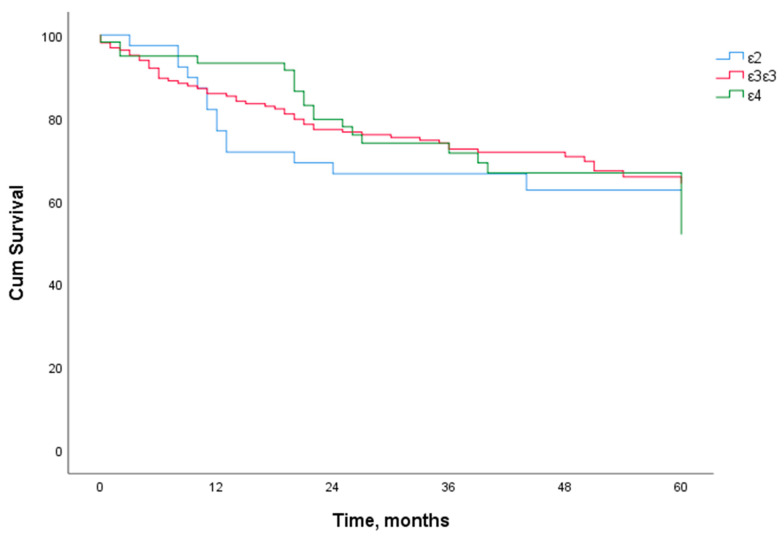
Five-year specific disease survival analysis of LSCC patients according to *ApoE* genotype and alleles (ε2, ε3ε3, and ε4).

**Table 1 biomolecules-12-01013-t001:** Demographic characteristics of the study groups.

Characteristics		Group		*p*-Value
		LSCC (n = 291)	Control Group (n = 311)	
Males, n (%)		282 (96.9)	301 (96.8)	0.931 *
Females, n (%)		9 (3.1)	10 (3.2)	
Age, median (IQR ***)		64 (9)	66 (9)	0.054 **
Stage, n (%)	I	104 (35.7)	-	-
	II	64 (22.0)		
	III	55 (18.9)		
	IV	68 (23.4)		
T, n (%)	1	107 (36.8)	-	-
	2	60 (20.6)		
	3	61 (21.0)		
	4	63 (21.6)		
N, n (%)	0	245 (84.2)	-	-
	1	16 (5.5)		
	2	30 (10.3)		
M, n (%)	0	288 (99.0)	-	-
	1	3 (1.0)		
G, n (%)	1	86 (29.5)	-	-
	2	180 (61.9)		
	3	25 (8.6)		

* Pearson Chi-square test; ** Mann–Whitney U test; *** interquartile range. LSCC—laryngeal squamous cell carcinoma; T—tumor size; M—metastasis; N—metastasis to neck lymph nodes; G—tumor differentiation grade.

**Table 2 biomolecules-12-01013-t002:** Distribution of *ApoE* genotypes and alleles between patients with LSCC and control group.

Gene, Genotype	LSCC * (n = 291)	Control Group (n = 311)	*p*-Value
*ApoE* ε2/ε2, n (%)	3 (1.0)	2 (0.6)	0.600
*ApoE* ε2/ε3, n (%)	39 (13.4)	47 (15.1)	0.549
*ApoE* ε2/ε4, n (%)	5 (1.7)	15 (4.8)	**0.034**
*ApoE* ε3/ε3, n (%)	178 (61.2)	150 (48.2)	**0.001**
*ApoE* ε3/ε4, n (%)	55 (18.9)	81 (26.0)	**0.036**
*ApoE* ε4/ε4, n (%)	11 (3.8)	16 (5.1)	0.439
ε2 allele, n (%)	50 (8.6)	66 (10.6)	0.239
ε3 allele, n (%)	450 (77.5)	428 (68.8)	**<0.001**
ε4 allele, n (%)	81 (13.9)	128 (20.6)	**0.002**

* OLSCC—laryngeal squamous cell carcinoma.

**Table 3 biomolecules-12-01013-t003:** Binary logistic regression analysis of *ApoE*.

Genotype	OR * (95% CI **)	*p*-Value ***
*ApoE* ε2/ε2	1.609 (0.267–9.701)	0.604
*ApoE* ε2/ε3	0.869 (0.550–1.375)	0.549
*ApoE* ε2/ε4	0.345 (0.124–0.962)	**0.042**
*ApoE* ε3/ε3	1.691 (1.223–2.338)	**0.001**
*ApoE* ε3/ε4	0.662 (0.449–0.975)	**0.037**
*ApoE* ε4/ε4	0.724 (0.330–1.588)	0.421

* OR—odds ratio; ** CI—confidence interval; *** *p* value—significance level (alpha = 0.05).

**Table 4 biomolecules-12-01013-t004:** Distribution of *ApoE* genotypes and alleles frequencies between the control and patients with LSCC (in early and advanced stages) groups.

Gene, Genotype	Control Group (n = 311)	Early-Stage LSCC * (n = 168)	*p*-Value	Advanced Stage LSCC (n = 123)	*p*-Value
*ApoE* ε2/ε2, n (%)	2 (0.6)	2 (1.2)	0.530	1 (0.8)	0.847
*ApoE* ε2/ε3, n (%)	47 (15.1)	21 (12.5)	0.434	18 (14.6)	0.890
*ApoE* ε2/ε4, n (%)	15 (4.8)	5 (3.0)	0.335	0 (0.0)	-
*ApoE* ε3/ε3, n (%)	150 (48.2)	101 (60.1)	**0.013**	77 (62.6)	**0.007**
*ApoE* ε3/ε4, n (%)	81 (26.0)	32 (19.0)	0.085	23 (18.7)	0.106
*ApoE* ε4/ε4, n (%)	16 (5.1)	7 (4.2)	0.633	4 (3.3)	0.397
ε2 allele, n (%)	66 (10.6)	30 (8.9)	0.408	20 (9.0)	0.486
ε3 allele, n (%)	428 (68.8)	255 (75.9)	**0.021**	172 (77.1)	**0.019**
ε4 allele, n (%)	128 (20.6)	51 (15.2)	**0.041**	31 (13.9)	**0.029**

* LSCC—laryngeal squamous cell carcinoma.

**Table 5 biomolecules-12-01013-t005:** Binary logistic regression analysis of *ApoE* in the control group and early and advanced LSCC stage subgroups.

LSCC *	Gene, Genotype	OR ** (95% CI ***)	*p*-Value ****
Early stage	*ApoE* ε2/ε2	1.861 (0.260–13.335)	0.536
	*ApoE* ε2/ε3	0.802 (0.462–1.394)	0.435
	*ApoE* ε2/ε4	0.605 (0.216–1.696)	0.339
	*ApoE* ε3/ε3	1.618 (1.106–2.367)	**0.013**
	*ApoE* ε3/ε4	0.668 (0.421–1.059)	0.086
	*ApoE* ε4/ε4	0.802 (0.323–1.989)	0.633
Advanced stage	*ApoE* ε2/ε2	0.848 (1.266–14.094)	0.848
	*ApoE* ε2/ε3	0.963 (0.535–1.734)	0.900
	*ApoE* ε2/ε4	-	-
	*ApoE* ε3/ε3	1.797 (1.172–2.755)	**0.007**
	*ApoE* ε3/ε4	0.653 (0.389–1.098)	0.108
	*ApoE* ε4/ε4	0.620 (0.203–1.892)	0.401

* LSCC—laryngeal squamous cell carcinoma; ** OR—odds ratio; *** CI—confidence interval; **** *p* value—significance level (alpha = 0.05).

**Table 6 biomolecules-12-01013-t006:** Frequencies of *ApoE* genotypes and alleles in the healthy control group and LSCC patient subgroups regarding tumor size.

Gene, Genotype	Control Group (n = 311)	T1 (n = 107)	*p*-Value	T2 (n = 60)	*p*-Value	T3 (n = 61)	*p*-Value	T4 (n = 63)	*p*-Value
*ApoE* ε2/ε2, n (%)	2 (0.6)	1 (0.9)	0.758	0 (0.0)	-	1 (1.6)	0.426	1 (1.6)	0.444
*ApoE* ε2/ε3, n (%)	47 (15.1)	15 (14.0)	0.784	6 (10.0)	0.300	9 (14.8)	0.943	9 (14.3)	0.847
*ApoE* ε2/ε4, n (%)	15 (4.8)	5 (4.7)	0.950	0 (0.0)	-	0 (0.0)	-	0 (0.0)	-
*ApoE* ε3/ε3, n (%)	150 (48.2)	66 (61.7)	**0.016**	35 (58.3)	0.152	37 (60.7)	0.076	40 (63.5)	**0.027**
*ApoE* ε3/ε4, n (%)	81 (26.0)	15 (14.0)	**0.011**	17 (28.3)	0.713	12 (19.7)	0.293	11 (17.5)	0.149
*ApoE* ε4/ε4, n (%)	16 (5.1)	5 (4.7)	0.847	2 (3.3)	0.550	2 (3.3)	0.535	2 (3.2)	0.505
ε2 allele, n (%)	66 (10.6)	22 (10.3)	0.892	6 (5.0)	0.057	11 (9.0)	0.597	11 (8.7)	0.526
ε3 allele, n (%)	428 (68.8)	162 (75.7)	0.056	93 (77.5)	0.057	95 (77.9)	**0.045**	100 (79.4)	**0.018**
ε4 allele, n (%)	128 (20.6)	30 (14.0)	**0.034**	21 (17.5)	0.441	16 (13.1)	0.056	15 (11.9)	**0.024**

**Table 7 biomolecules-12-01013-t007:** Binary logistic regression analysis of *ApoE* in the control group and patients with LSCC in different tumor size subgroups.

Tumor Size	Gene, Genotype	OR * (95% CI **)	*p*-Value
T1	*ApoE* ε2/ε2	1.458 (0.131–16.238)	0.759
	*ApoE* ε2/ε3	0.916 (0.489–1.716)	0.784
	*ApoE* ε2/ε4	0.967 (0.343–2.728)	0.950
	*ApoE* ε3/ε3	1.728 (1.103–2.706)	**0.017**
	*ApoE* ε3/ε4	0.463 (0.254–0.845)	**0.012**
	*ApoE* ε4/ε4	0.904 (0.323–2.529)	0.847
T2	*ApoE* ε2/ε2	-	-
	*ApoE* ε2/ε3	0.624 (0.254–1.533)	0.304
	*ApoE* ε2/ε4	-	-
	*ApoE* ε3/ε3	1.503 (0.859–2.629)	0.154
	*ApoE* ε3/ε4	1.123 (0.606–2.078)	0.713
	*ApoE* ε4/ε4	0.636 (0.142–2.840)	0.553
T3	*ApoE* ε2/ε2	2.575 (0.230–28.852)	0.443
	*ApoE* ε2/ε3	0.943 (0.449–2.105)	0.943
	*ApoE* ε2/ε4	-	-
	*ApoE* ε3/ε3	1.655 (0.945–2.896)	0.078
	*ApoE* ε3/ε4	0.695 (0.352–1.373)	0.295
	*ApoE* ε4/ε4	0.625 (0.140–2.791)	0.538
T4	*ApoE* ε2/ε2	2.492 (0.222–27.909)	0.459
	*ApoE* ε2/ε3	0.936 (0.434–2.024)	0.867
	*ApoE* ε2/ε4	-	-
	*ApoE* ε3/ε3	1.867 (1.067–3.267)	**0.029**
	*ApoE* ε3/ε4	0.601 (0.299–1.207)	0.152
	*ApoE* ε4/ε4	0.605 (0.135–2.697)	0.509

* OR—odds ratio; ** CI—confidence interval.

**Table 8 biomolecules-12-01013-t008:** Frequencies of *ApoE* genotypes and alleles in healthy controls and LSCC patients with (N1–2) and without (N0) metastases to the neck lymph nodes.

Gene, Genotype	Control Group (n = 311)	LSCC * (N0) (n = 245)	*p*-Value	LSCC (N1–2) (n = 46)	*p*-Value
*ApoE* ε2/ε2, n (%)	2 (0.6)	1 (0.4)	0.707	2 (4.3)	**0.026**
*ApoE* ε2/ε3, n (%)	47 (15.1)	31 (12.7)	0.407	8 (17.4)	0.689
*ApoE* ε2/ε4, n (%)	15 (4.8)	5 (2.0)	0.080	0 (0.0)	-
*ApoE* ε3/ε3, n (%)	150 (48.2)	150 (61.2)	**0.002**	28 (60.9)	0.110
*ApoE* ε3/ε4, n (%)	81 (26.0)	48 (19.6)	0.074	7 (15.2)	0.112
*ApoE* ε4/ε4, n (%)	16 (5.1)	10 (4.1)	0.556	1 (2.2)	0.377
ε2 allele, n (%)	66 (10.6)	39 (7.9)	0.131	12 (13.0)	0.485
ε3 allele, n (%)	428 (68.8)	379 (77.2)	**0.002**	71 (77.2)	0.103
ε4 allele, n (%)	128 (20.6)	73 (14.9)	**0.014**	9 (9.8)	**0.014**

* LSCC—laryngeal squamous cell carcinoma; N0—no metastases to the neck lymph nodes.

**Table 9 biomolecules-12-01013-t009:** Binary logistic regression analysis of *ApoE* in the control group and LSCC patients with (N1–2) and without (N0) metastases to the lymph nodes.

LSCC *	Gene, Genotype	OR ** (95% CI ***)	*p*-Value
**N0**	*ApoE* ε2/ε2	0.633 (0.057–7.024)	0.710
	*ApoE* ε2/ε3	0.814 (0.499–1.326)	0.408
	*ApoE* ε2/ε4	0.411 (0.147–1.147)	0.090
	*ApoE* ε3/ε3	1.695 (1.206–2.381)	**0.002**
	*ApoE* ε3/ε4	0.692 (0.462–1.037)	0.074
	*ApoE* ε4/ε4	0.785 (0.350–1.761)	0.556
**N1–2**	*ApoE* ε2/ε2	7.023 (0.965–51.131)	0.054
	*ApoE* ε2/ε3	1.183 (0.519–2.693)	0.690
	*ApoE* ε2/ε4	-	-
	*ApoE* ε3/ε3	1.670 (0.887–3.143)	0.112
	*ApoE* ε3/ε4	0.510 (0.219–1.185)	0.117
	*ApoE* ε4/ε4	0.410 (0.053–3.165)	0.392

* LSCC—laryngeal squamous cell carcinoma; ** OR—odds ratio; *** CI—confidence interval; N0—no metastases to the neck lymph nodes; N1–2—with metastases ton the neck lymph nodes.

**Table 10 biomolecules-12-01013-t010:** Frequencies of *ApoE* genotypes and alleles in the healthy control group and LSCC patient subgroups with different tumor differentiation grades.

Gene, Genotype	Control Group (n = 311)	Well-Differentiated (n = 85)	*p*-Value	Poorly Differentiated (n = 205)	*p*-Value
*ApoE* ε2/ε2, n (%)	2 (0.6)	1 (1.2)	0.615	2 (1.0)	0.340
*ApoE* ε2/ε3, n (%)	47 (15.1)	13 (15.3)	0.967	26 (12.7)	0.438
*ApoE* ε2/ε4, n (%)	15 (4.8)	2 (2.4)	0.319	3 (1.5)	**0.042**
*ApoE* ε3/ε3, n (%)	150 (48.2)	55 (64.7)	**0.007**	122 (59.5)	**0.012**
*ApoE* ε3/ε4, n (%)	81 (26.0)	11 (12.9)	**0.011**	44 (21.5)	0.235
*ApoE* ε4/ε4, n (%)	16 (5.1)	3 (3.5)	0.537	8 (3.9)	0.551
ε2 allele, n (%)	66 (10.6)	17 (10.0)	0.818	33 (8.0)	0.171
ε3 allele, n (%)	428 (68.8)	134 (78.8)	**0.011**	314 (76.6)	**0.007**
ε4 allele, n (%)	128 (20.6)	19 (11.2)	**0.005**	63 (15.4)	**0.035**

**Table 11 biomolecules-12-01013-t011:** Binary logistic regression analysis of *ApoE* in the control group and LSCC patient subgroups with different tumor differentiation grades.

Tumor Differentiation Grades	Gene, Genotype	OR * (95% CI **)	*p*-Value
Well-differentiated	*ApoE* ε2/ε2	1.839 (0.165–20.531)	0.621
	*ApoE* ε2/ε3	1.014 (0.520–1.976)	0.967
	*ApoE* ε2/ε4	0.476 (0.107–2.121)	0.330
	*ApoE* ε3/ε3	1.968 (1.197–3.236)	**0.008**
	*ApoE* ε3/ε4	0.422 (0.213–0.835)	**0.013**
	*ApoE* ε4/ε4	0.675 (0.192–2.371)	0.539
Poorly differentiated	*ApoE* ε2/ε2	1.522 (0.213–10.893)	0.676
	*ApoE* ε2/ε3	0.816 (0.487–1.366)	0.439
	*ApoE* ε2/ε4	0.293 (0.084–1.025)	0.055
	*ApoE* ε3/ε3	1.578 (1.104–2.254)	**0.012**
	*ApoE* ε3/ε4	0.776 (0.511–1.180)	0.235
	*ApoE* ε4/ε4	0.749 (0.314–1.783)	0.513

* OR—odds ratio; ** CI—confidence interval.

**Table 12 biomolecules-12-01013-t012:** Genotype distribution and serum ApoE levels.

Gene, Genotype, (n)	ApoE Level (ng/mL)		*p*-Value
	LSCC Means (SD)	Controls Means (SD)	
*ApoE* ε2/ε2, (0)	-	-	-
*ApoE* ε2/ε3, (5)	1.865 (0.470)	1.607 (0.073)	0.443 *
*ApoE* ε2/ε4, (0)	-	-	-
*ApoE* ε3/ε3, (14)	1.947 (0.284)	1.956 (0.289)	0.955 *
*ApoE* ε3/ε4, (14)	2.074 (0.534)	1.705 (0.419)	0.195 *
*ApoE* ε4/ε4, (5)	2.111 (0.624) *	-	N/A

* Student’s *t*-test.

## Data Availability

Not applicable.

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
