# Peer review of "The Role of ApoE Serum Levels and ApoE Gene Polymorphisms in Patients with Laryngeal Squamous Cell Carcinoma"

_biomolecules, 2022, doi:10.3390/biom12081013_

Round 1

Reviewer 1 Report

Dear authors,

Thanks for your great efforts in conducting a paper like yours!

Before acceptance some minor changes are mandatory:

@Introduction: Please clearify APOE as being the gene and describe its chromosomal location in humans. Furthermore, please describe "Apolipoprotein E" (ApoE) being the gene product, which is very important to the reader for further understanding!

Please describe all abbreviations (incl. CT, MRI, EDTA, and DNA) fully when mentioning first.

Alcohol itself is definitely NOT causing LSCC! But it is a potentialyzer of canerogenic smoking! Please correct this misunderstanding!

Additionally, it might be usefull when correlating survival to differentiate the different therpy regimens (surgery vs. irradiation). Would help the paper to reach for clinicians. I'd suggest to al least write a statement, or better to give a chart overview.

Thank you!

Author Response

Thank you very much for your positive evaluation of our study.

Reviewer 2 Report

A very welcome contribution to head and neck cancer biomarkers. However, there are a number of inaccuracies that must be corrected before the paper can be accepted for publication. This relates to most sections of the paper. Please find comments below.

M&M section requires some revision. Description of patient selection must be improved, providing information about time of enrollment, scheduling the retrospective analyses, obtaining blood for genotyping and serum for ELISA measurements etc.

As the authors stated in INTRODUCTION that LSCC caused over 120 deaths in Lithuania in 2020, how many years were required to obtain the LSCC sample at the single institution? Was there a change in distribution of T and N categories, stage etc. over time? 

2.3. Statistical analysis can be improved

The description can and must be improved. Take a look at other MDPI papers.

Table 1: Please set n and p italic. Asterisks should be reserved for labeling statistical significance; choose another sign for IQR.

Line 166 (Foot note of Table 2: ) I guess "LSCC vs. Control group p=0.018" reflects the summary statistics for comparing both groups, but this should be outlined more clearly.

Table 3 is missing

Line 175ff: The intention of the authors to demonstrate associations of  ApoE genotypes with clinical characteristics and contrasting early and advanced stages seems to be clear. However, comparing early and advanced stages bears the risk of some inaccuracies probably not intended by the authors. From a biological point of view, growth of the primary lesion alone or causing impaired function (fixation of vocal cord leading to T3 that will lead to staging at least as UICC III and therefore advanced stage) may not demonstrate the biological consequences as desired. The authors' aim, I guess, was to highlight the different frequencies of ε3/ε3 carriers in LSCC and the modifying impact of ApoE ε4/ε3 and ApoE ε4/ε4 vs. ε3/ε3 on biological differences (Tables 11 and 12). Biological differences might be found between patients with ε2 and ε4 related to local metastasis, and the for  ε4 expected lower frequency in N-positive LSCC indeed was found and is presented in Tables 9 and 10. However, supraglottic T1 and T2 vs. T3 and T4 LSCC may or may not be linked to biological differences, and maximum diameter of 3.8 cm (T2) vs. 4.1 cm (T3) after or  before formaldehyd fixation could be without any meaning.

Figure 1 should show serum concentrations of APOE. However, correct scaling is missing. What is shown on the Y-axis?

Table 13 did not show the correct units (should be ng/ml instead of pg/ml, right?) and provides no information of the distribution of numbers per genotype. There is also no information about how the 20 LSCC cases and 18 control patients were selected. I guess the samples were measured in duplicates instead of inflating number of cases and controls measured with the single available ELISA. Please explain.

What genotypes are ε1ε3 and ε1ε4?

Figure 2. This figure looks like a draft. Moreover, it is not clear to me why you have chosen to omit showing of survival data of cases with minor frequency. I suggest showing first a waterfall plot for survival of patients in genotype groups as Figure 2 first. 

Thereafter please show two much more clearer Kaplan-Meier cumulative survival plots contrasting "ε2 carrier",  "ε4 carrier" and  "ε3 homozygous carrier" for overall survival (OS) and disease-specific survival (DSS) in Figure 3A and 3B. Scaling of both KM plots should be corrected to adhere to standards, that is X-axis scaled in 12, 24, ... 60 months, Y-axis in percentage including the median line.

Discussion

It is not clear to me why the increased frequency of ε3ε3 is discussed a protective for LSCC. The opposite is true. 

It should be mentioned either in the Introduction or at least here in line 296 (and therefore before discussing nucleotide positions thereafter) that the rs7412(T) allele always indicates presence of ε2 allele (cys112, cys158), whereas the more common C is mostly found together with the common rs429358(T) allele and defines the ε3 allele (cys112, arg158). If the rs429358 allele is (C) and the same chromosome harbors the rs7412(C) allele, the combination is known as an APOE-ε4 allele (arg112, arg158).

There is insufficient data  from only 20 ApoE ELISA measurements to state in line 362ff "No correlation was identified between ApoE protein levels and the 5-year survival rate of patients with LSCC. No effect of ApoE SNPs on the 5-year survival rate of these patients was revealed". Taking into account the unknown patient selection and few measurements derived from those few patients at a not specified point of time. Before or during treatment? Storage conditions? This needs to be provided in the M&M section.

Discussing strength of the study beginning in line 365ff contains many overstatements. There is no reason to do that. The study done in LSCC has the advantage to provide information about larynx cancer patients but also these are distinct according to localization, subglottic vs. glottic vs. supraglottic, with and without neck nodes, being related to smoking or not, occupational exposure, age,...

The paper discusses rather reports about impact of ApoE alleles on other cancer entities, and Alzheimers

The following conclusions are hard to accept:

Conclusions

ApoE ε2/ε4 and ε3/ε4 were found to play a significant role in LSCC development, while ApoE ε3/ε3 may have a protection position in LSCC development.

The frequency of ApoE ε2/ε4 and ε3/ε4 were reported to be lower in LSCC compared to Controls (but not significant !), whereas ApoE ε3/ε3 is significantly increased in frequency (this means, poses a risk for development of LSCC!).  As the frequency of ε4 carriers is lower in LSCC, there must be a protective role of ε4 for developing LSCC! It is difficult to understand how such a sentence came into the paper able to destroy an otherwise good report. Lower frequencies compared to controls should always reflect a protective effect for developing the disease, and obviously the reduced frequencies of ε4 carriers in line with improved outcome of ε4 carriers (according to your Kaplan-Meier plot at least up to three years, if yellow, blue and orange are combined) are indicating that both a reduced odds ratio and a (not significant, but numerical) reduced hazard ratio are found. Therefore, your study's results argue for protective effects of ε4 in LSCC. Please do not destroy your very welcome contribution to the field by failing to draw the appropriate conclusions!

This, however, means major revision taking into account the above made suggestions should substantially improve the paper and potentially will make it acceptable for publication. 

Author Response

(The authors gave the same response as above.)

Round 2

Reviewer 2 Report

Many thanks for revising the paper appropriately. There are only a few orthographic mistakes remaining that could be quickly corrected while preparing the files proofs. The paper is now acceptable for publication.

Author Response

Thanks